# Roles of mothers and fathers in supporting child physical activity: a cross-sectional mixed-methods study

Emma Solomon-Moore,[1] Zoi Toumpakari,[1] Simon J Sebire,[1] Janice L Thompson,[2] Deborah A Lawlor,[3,4] Russell Jago[1]

[1]Centre for Exercise, Nutrition and Health Sciences, School for Policy Studies, University of Bristol, Bristol, UK
[2]School of Sport, Exercise and Rehabilitation Sciences, University of Birmingham, Birmingham, UK
[3]MRC Integrative Epidemiology Unit, University of Bristol, Bristol, UK
[4]Population Health Sciences, Bristol Medical School, University of Bristol, Bristol, UK

**Correspondence to**
Dr Russell Jago;
russ.jago@bristol.ac.uk

## ABSTRACT

**Objectives** Examine the extent that parent gender is associated with supporting children's physical activity.

**Design** Cross-sectional mixed-methods study.

**Setting** 47 primary schools located in Bristol (UK).

**Participants** 944 children aged 8–9 years and one of their parents provided quantitative data; 51 parents (20 fathers) were interviewed.

**Methods** Children wore an accelerometer, and mean minutes of moderate-to-vigorous physical activity (MVPA) per day, counts per minute (CPM) and achievement of national MVPA guidelines were derived. Parents reported who leads in supporting child activity during the week and weekend. Linear and logistic regression examined the association between gender of parent who supports child activity and child physical activity. For the semistructured telephone interviews, inductive and deductive content analyses were used to explore the role of gender in how parents support child activity.

**Results** Parents appeared to have a stronger role in supporting boys to be more active, than girls, and the strongest associations were when they reported that both parents had equal roles in supporting their child. For example, compared with the reference of female/mother support, equal contribution from both parents during the week was associated with boys doing 5.9 (95% CI 1.2 to 10.6) more minutes of MVPA per day and more CPM when both parents support on weekday and weekends (55.1 (14.3 to 95.9) and 52.8 (1.8 to 103.7), respectively). Associations in girls were weaker and sometimes in the opposite direction, but there was no strong statistical evidence for gender interactions. Themes emerged from the qualitative data, specifically; parents proactively supporting physical activity equally, mothers supporting during the week, families getting together at weekends, families doing activities separately due to preferences and parents using activities to bond one-to-one with children.

**Conclusions** Mothers primarily support child activity during the week. Children, possibly more so boys, are more active if both parents share the supporting role.

## INTRODUCTION

Children who are physically active are at a lower risk of obesity, high blood pressure, metabolic syndrome and depression.[1 2] The UK Government recommends that children and young people aged 5–18 years should

### Strengths and limitations of this study

► Mixed-methods study.
► Accelerometer data from a large sample of children aged 8–9 years.
► Semistructured telephone interviews with 51 parents, including 20 fathers.
► Cross-sectional study design from a single UK region.
► The measurement of parental support of child physical activity would be strengthened by collecting data from both parents and information on the quality and quantity of support.

engage in at least 60 min of moderate-to-vigorous physical activity (MVPA) every day.[3] However, data from the nationally representative Millennium cohort showed that only 51% of children aged 7–8 years met the recommendation.[4] Physical activity declines throughout childhood and adolescence, with boys being more active than girls at all ages.[4–9] Thus, in order to develop effective means of increasing child physical activity, there is a need to understand the factors that influence behaviour.

Parents act as gatekeepers to children's activity[10] and can play an important role in increasing their child's physical activity.[11–13] For instance, parents can influence their child's activity by being active with their child, role-modelling active behaviour and/or by facilitating physical activity for their child (logistic support).[13–16] Studies examining associations between parent and child physical activity behaviour have yielded mixed results.[14 17–20] A growing body of research has shown that providing logistic support is associated with increased physical activity[21–23] and, therefore, may be the most important source of parental influence on children's activity.

The gender of the parent who takes the lead in supporting child activity could be an

important influence on children's activity levels. Traditional gender roles comprised the public sphere (employment, education and politics) being dominated by men and the private sphere (home and family) being exclusively the realm of women.[24] However, these traditional roles have been shifting, as explained by the gender revolution framework,[25] whereby men's attitudes have become much more accepting of gender equality in the family,[26] particularly in caring for children.[27] It is not clear what the current role gender plays in parental physical activity support. Several studies suggest that mothers play a larger role in the logistical planning of children's physical activity, whereas fathers are more likely to model physical activity.[28 29] However, most studies in this area have focused on the mother–child relationship, and relatively little attention has been paid to the role of fathers.[30] From qualitative interviews with parents of children aged 5–6 years in the B-PROACT1V study, we found evidence that fathers play a key role in promoting children's physical activity, influencing their choices and behaviours,[31] a finding replicated in other studies.[32 33] The Healthy Dads, Healthy Kids intervention demonstrated that engaging fathers in physical activity with their children can promote increased physical activity among children.[34 35] Data from B-PROACT1V interviews suggest that fathers may take more responsibility for their son's physical activity (eg, taking their son to sports clubs) and mothers with their daughter's activity.[31] To date, there is inconsistent evidence regarding whether gender-specific parental influence (ie, mothers with daughters and fathers with sons) is stronger than cross-gender parental influence (ie, mothers with sons and fathers with daughters) on children's physical activity.[28 36–39] Therefore, a greater understanding is needed about the role gender plays in how parents support their child to be active, and if this varies by child gender.

The aim of this mixed-methods study was to examine parent gender, in terms of which parent supports their child to be active and its association with child physical activity. A secondary aim was to discover if these associations varied by child gender.

## METHODS

Data are from the longitudinal B-PROACT1V study, which aimed to examine factors associated with children's and parents' physical activity, sedentary time and screen-viewing behaviours. The study has been described in detail elsewhere.[9 17 40] Briefly, in 2012 and 2013, data were collected from 1299 year 1 children (5–6 years) from 57 primary schools across Bristol, UK. Between March 2015 and July 2016, 47 of the original schools were re-recruited, and data were collected from 1223 year 4 children (8–9 years). One of the children's parents was also recruited to the study. The current study used a mixed-methods design, incorporating cross-sectional data from the year 4 assessments, for the 944 children and parents who provided valid child accelerometer data and complete parent questionnaire data for questions on child and parent demographics and gender roles associated with supporting child activity (figure 1), with qualitative data via semistructured telephone interviews from a subsample of 51 parents (details below; figure 2). The current study incorporated a convergent parallel mixed-methods design. Quantitative data were collected prior to qualitative data collection, but the analyses and interpretation

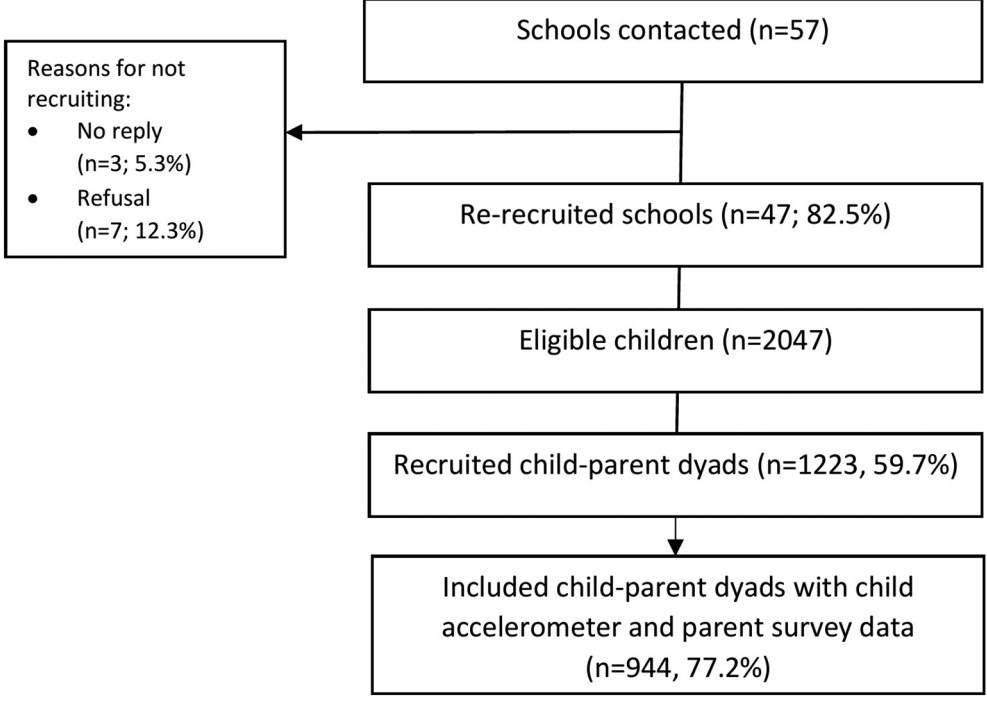

**Figure 1** Study flow of participants for the quantitative study.

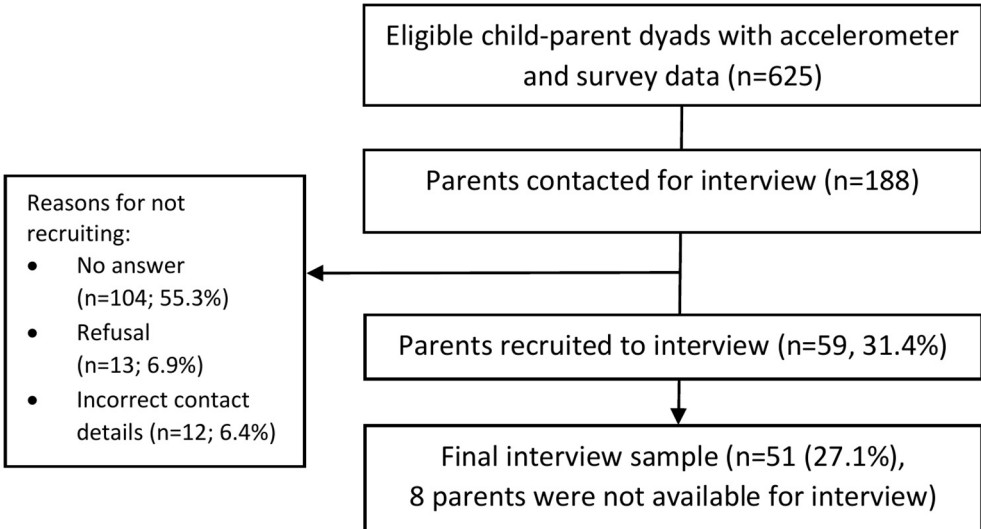

**Figure 2** Study flow of participants for the qualitative study.

were conducted in parallel.[41] Written parent consent was received for all participants.[42]

## Accelerometer data

Children wore a waist-worn ActiGraph wGT3X-BT accelerometer for 5 days including two weekend days. Waist-worn accelerometers have been demonstrated to be valid for measuring physical activity in children.[43 44] Accelerometer data were processed using Kinesoft (V.3.3.75; Kinesoft, Saskatchewan, Canada), and were included in the primary analyses if children provided at least 3 days of valid data (including at least one weekend day). A valid day was defined as at least 500 min of data after excluding intervals of ≥60 min of zero counts, allowing up to 2 min of interruptions. Minutes spent in MVPA were derived using population-specific cut points for children.[45] In a comparative study with other widely used accelerometer cut points, the Evenson thresholds[45] (in which stair climbing and brisk walking corresponded to MVPA) were shown to provide the most accurate assessments of children's energy expenditure.[46] Mean accelerometer counts per minute (CPM) and a binary variable indicating whether the child's average daily MVPA was greater than the 60 min per day recommended by the UK government[3] were also derived.

## Parent support variables

To understand the gender roles associated with parents supporting their child's activity, parents were asked three questions via a questionnaire: (a) "In your family who takes the lead role in supporting your Year 4 child to be active during the week?", (b) "In your family who takes the lead role in supporting your Year 4 child to be active at the weekend?" and (c) "Who do you think should take the lead role in supporting your Year 4 child to be active?". Each question had three response options: 'Mother/ Female caregiver', 'Father/Male caregiver' or 'About the same' for questions (a) and (b) and 'Should be shared' for question (c).

## Demographic information

Parents provided demographic information via a questionnaire, including parent and child gender, date of birth and ethnic origin. Where children's date of birth was missing (21% of children), they were assigned the median age of 9.0 years (as the children were all in the same school year with a maximum age difference between the youngest and oldest of just under 12 months legally possible). As an indicator of socioeconomic status, Index of Multiple Deprivation (IMD) scores, based on the English Indices of Deprivation,[47] were assigned to each child based on their reported home postcode, where higher scores indicate greater levels of deprivation. IMD scores provide a set of relative measures of deprivation for lower-layer super output areas across England, based on seven different domains of deprivation: income deprivation; employment deprivation; education, skills and training deprivation; health deprivation and disability; crime; barriers to housing and services and living environment deprivation. Child height, weight and blood pressure were also measured.

## Interview data

During consent procedures, parents were informed that they may be recontacted to take part in a telephone interview. Only families with complete data for all measures (accelerometer and questionnaire data, child height, weight and blood pressure) were included in the interview sample (n=625, of which 161 (25.8%) had data from fathers). This sample was stratified according to the child's MVPA minutes per day (dichotomised around the study median: 57.5 min), sedentary minutes per day (dichotomised around the median: 434.6 min) and child gender. This produced eight subgroups (1=low MVPA, low sedentary time boys and 8=high MVPA, high sedentary time girls; online supplementary table S1). The order in which parents were invited to participate in an interview was randomised within each subgroup. Contact

attempts were made with 188 parents in total, of which 59 (31.4%) initially agreed to participate in an interview, and 51 (27.1%) completed an interview (figure 2). Interviews were audio-recorded and continued until theoretical saturation was reached for the entire sample and the subgroups. Parents were invited to participate by telephone between July and October 2016, and interviews were conducted at the interviewee's convenience (37 during weekday daytimes (72.5%), 13 during weekday evenings (25.5%) and 1 on a weekend evening (2%)). Participants were sent a £10 high street shopping voucher as a thank you for their time.

An interview guide was developed and refined by the research team based on identifying gaps in current knowledge and guided by the year 1 B-PROACT1V quantitative and qualitative findings. This included questions relating to a variety of topics, including parents' perceptions of their child's physical activity and screen-viewing behaviours,[48] strategies for managing these behaviours,[49 50] understanding what has changed regarding these behaviours[17 40] and understanding how family dynamics influence children's physical activity.[51] The need to engage more fathers in research was also identified as a priority.[31 51] Questions were posed in a non-leading manner to allow participants to shape the direction of the interview, and issues that emerged were probed. Interviews were conducted by two female researchers (qualified to at least MSc level) who were trained in conducting qualitative interviews.

### Data analysis
#### Quantitative data
Means, proportions and $X^2$ statistics were used to examine the distributions of exposures, outcomes and covariates between participants included and excluded in this study and between child and parent gender. Nearly all parents reported that both parents 'should take the lead' in supporting their child's activity (93.8%); therefore, we could not explore the association of parental attitudes towards who should support child physical activity, as numbers were too small in the mother or father only categories. We used linear regression models to examine the associations of parent support of child activity during the week and weekend with the child's MVPA minutes per day and CPM and logistic regression models to examine associations with achievement of the MVPA guideline. Models were adjusted for child age, gender of parent providing the information on support and household IMD score. Robust standard errors were used to account for the clustering of children in schools for all models. Models were examined for all children and separately for boys and girls. Combined Wald tests were used to test for evidence of interaction between child gender and the exposure of interest. All analyses were performed in Stata V.14.0 (StataCorp, 2015).

#### Qualitative data
Interviews were transcribed verbatim and anonymised before being entered into QSR NVivo 10 (QSR

International, Warrington, UK) to facilitate analysis. Using the framework method, thematic content analysis was performed by two researchers, enabling themes to develop both inductively from the accounts (experiences and views) of participants and deductively from existing literature.[52 53] Analysis involved several phases: familiarisation, coding, developing a framework, applying the framework, charting data into the framework matrix and interpretation. During familiarisation, transcripts were thoroughly read and re-read independently by two researchers to immerse themselves in the data. After discussion between the two researchers, an initial coding frame was developed and applied to the data based on pre-existing ideas and was refined throughout the process to allow for the inductive emergence of additional themes. The two researchers met regularly to ensure accuracy and consistency. Any disagreements that occurred during coding were discussed with additional members of the research team to ensure consensus, and no disagreements remained unsolved. Hierarchies of categories were created and summarised, and brief summaries, mind maps and representative quotes for each category were abstracted for reporting purposes. The final quotes were selected as they are illustrative of several responses given by parents.

### RESULTS
#### Participant characteristics
The characteristics of the participants included and excluded from the quantitative dataset, and from the subset of interview participants, are shown in table 1. Of the 944 included families, the majority (680 (72%)) had data from a mother/female caregiver, with 264 (28%) from fathers/male caregivers. Children excluded due to missing data were more likely to be deprived and did less minutes of MVPA per day, but were otherwise similar to the included dataset. Of the interview participants (n=51), 31 were mothers, and 20 were fathers, with an average age of 41.2 (SD: 4.5) years, and 94.1% were white British. The interview participants were generally comparable to the main dataset, but tended to be less deprived. Interview participants were also more likely to be fathers and have less active children compared with the main dataset. The average interview duration was 34.4 min (SD 8.0 min, range: 18 to 55 min).

Online supplementary table S2 shows the gender of the parent who reportedly supports child physical activity by parent and child gender. Mothers reported that typically they led in supporting their child's physical activity during the week, whereas fathers generally reported that duties were shared between parents. Most mothers and fathers reported that both parents shared the role of supporting their child's activity at the weekend; however, 31% of mothers and 27% of fathers, respectively, reported that they led child activity.

The interview data generally supported this, with several mothers stating that they support their child to

**Table 1** Descriptive characteristics of the main study sample (n=944) and subset of interview participants (n=51)

| Characteristic | Included (n=944) Mean (SD) or % | Excluded N | Mean (SD) or % | P value | Interview sample (n=51) Mean (SD) or % |
|---|---|---|---|---|---|
| Child MVPA (min/day) | 62.8 (22.8) | 209 | 58.6 (21.4) | 0.01 | 58.3 (17.4) |
| Accelerometer counts per minute | 620.4 (203.2) | 209 | 609.0 (208.8) | 0.46 | 573.2 (142.0) |
| Met MVPA guidelines (≥ 60 min/day) | | 209 | | 0.06 | |
| No | 52.0 | | 59.3 | | 58.8 |
| Yes | 48.0 | | 40.7 | | 41.2 |
| Child gender | | 279 | | 0.73 | |
| Boy | 45.2 | | 46.4 | | 49.0 |
| Girl | 54.8 | | 53.6 | | 51.0 |
| Age of child (years) | 9.03 (0.46) | 279 | 9.04 (0.49) | 0.91 | 8.95 (0.37) |
| Household IMD* score | 15.1 (13.6) | 248 | 18.8 (15.5) | <0.001 | 11.5 (9.7) |
| Supports child activity during the week | | 39 | | 0.92 | |
| Mother | 48.8 | | 48.7 | | 43.1 |
| Father | 6.8 | | 5.1 | | 9.8 |
| Both parents | 44.4 | | 46.2 | | 47.1 |
| Supports child activity at the weekend | | 37 | | 0.35 | |
| Mother | 24.5 | | 32.4 | | 23.5 |
| Father | 17.7 | | 21.6 | | 23.5 |
| Both parents | 57.8 | | 45.9 | | 52.9 |
| Who should support child PA | | 38 | | 0.64 | |
| Mother | 5.2 | | 2.6 | | 3.9 |
| Father | 1.0 | | 0.0 | | 3.9 |
| Both parents | 93.8 | | 97.4 | | 92.2 |
| Parent gender | | 41 | | 0.24 | |
| Male | 28.0 | | 19.5 | | 39.2 |
| Female | 72.0 | | 80.5 | | 60.8 |
| Parent ethnic origin | | 53 | | 0.52 | |
| White British | 89.2 | | 91.3 | | 94.1 |

*A higher value indicates greater deprivation.
IMD, Index of Multiple Deprivation; MVPA, moderate-to-vigorous physical activity; PA, physical activity.

be active during the week out of necessity because fathers were working long hours or late into the evening. Some mothers also reported that they try to get the whole family together to do activities at the weekend, although this isn't always the norm.

On a weekday it's just, you know, every night we've got one or the other [children] have got a club on so it's just finish school and then me taking the children to their various clubs and then coming home and it's, erm, you know, pretty much get ready for bedtime… Weekends, yeah, we try to do stuff as a family. [Int 14, mother, girl, 63 MVPA min/day, mother supports weekday PA, both parents support weekend PA]

We like to do things as a family when we can; it's just all being around. My husband works quite late hours and things like that… He's, he's home when they're

going to bed usually … but like last Sunday, we all went swimming together as a family thing… but that isn't—to be honest, that isn't like, isn't like we would do that every weekend or anything [Int 35, mother, girl, 72 MVPA min/day, mother supports weekday PA, both parents support weekend PA]

Some parents indicated that they share the responsibility of supporting child physical activity, due to sharing an appreciation for the benefits of physical activity or because they value physical activity and feel a moral responsibility to fit activity in to the realities of life.

I'm active, my husband's active. And so, you know, we cascade that if you like down to the children so we, we don't really sit around at all, we're very active and on the go… [Int 3, mother, son, 59 MVPA

min/day, both parents support weekday and weekend PA]

Actively we are trying to get the children involved in the various, activities like where there's after-school or a swimming lesson or they are going to join Scouts, which will be helpful for them in the long run… So, so we, we are encouraging them to get involved in outdoor activities as much as possible. [Int 1, father, son, 76 MVPA min/day, both parents support weekday and weekend PA]

So wherever we can we'll always try and do the right thing [physical activity] and, you know, sometimes if it's not taking the car and it's walking distance we'll try and walk, and things like that… [Int 18, father, son, 86 MVPA min/day, father supports weekday and weekend PA]

A few parents reported sharing the responsibility of supporting child physical activity, but also doing activities separately due to child preferences. Examples included fathers and sons using physical activity time to bond over shared interests, while also giving mothers a respite for some 'me time', or parents taking children to separate activities to appease child preferences, avoid conflict and/or facilitate parent–child one-on-one time irrespective of gender.

We like going about walking as a family. Well, I say me and my husband do and we drag the kids along, but, you know, it's just getting some fresh air, but the boys have their own interests as well, such as the rugby or football which my husband takes the boys to. I have a bit of 'me time' when they go off to do that so, you know, it's a mix, I think. [Int 32, mother, girl, 86 MVPA min/day, both parents support weekday and weekend PA]

I would like to do a little bit more with them but because my son doesn't like what [child] likes and I would like to take them swimming together a little bit more so we can all go and do swimming but because he doesn't like it; we kind of end up two of us doing it and two of us not doing it [Int 29, mother, girl, 56 MVPA min/day, both parents support weekday and weekend PA]

I've said I might take him mountain biking this Sunday because I see that as exercise for him but also one to one. So, he's getting that, the benefit of obviously exercise, the sport that he actually really loves and is getting one to one time with a parent where, you know, it's hard isn't it, when there's other siblings [Int 3, mother, son, 59 MVPA min/day, both parents support weekday and weekend PA]

In the quantitative dataset, parents of girls tended to report that mothers take the lead in supporting their daughter's activity during the week, whereas parents of boys tended to report that the role was shared between both parents. Parents of boys and girls generally reported that they shared the responsibility of supporting child activity at the weekend, although parents of girls were more likely to report that mothers supported their daughter's weekend activity.

In contrast, the interview data revealed a mix of gender patterns associated with supporting child physical activity, not just mothers supporting daughters and fathers supporting sons. Some fathers reported that they supported their daughter's physical activity through chauffeuring them to sports clubs and expressed that they do so not just for logistical reasons, but also because they get real enjoyment from watching. A few mothers reported a lack of confidence in their own physical activity, because they aren't 'naturally sporty' and so they tend to let fathers take the lead in supporting child physical activity.

Yeah, she's been playing football for two and a half seasons now… and she's passionate about that. So I'm just a sort of chauffeur dad… that stands on the touchline in the cold windy rain. I enjoy that. [Int 51, father, girl, 71 MVPA min/day, father supports weekday and weekend PA]

Not that confident cause, like I say, I'm not actually naturally sporty or active. So it would be something that we would probably do as a family with their dad, and we could do it together… He's more confident, yeah, and he's more knowledgeable really with all that kind of stuff. And he's a—and he's the kind of person that's very much into, 'Come on, let's give it a go. Let's try and see. We might really enjoy it,' whereas I'm a bit more like, 'Oh no, don't make me do this. I'm really nervous.' And so I would probably shy away from it. [Int 24, mother, girl, 43 MVPA min/day, mother supports weekday PA, father supports weekend PA]

### Associations of who supports child activity with child physical activity variables

Table 2 shows the mean difference in child MVPA minutes per day by which parent/s take the lead in supporting child activity during the week and weekend. Compared with reporting that mothers support child activity (reference group), reporting that parents share the role of supporting child activity during the week was associated with children doing, on average, an additional 3.5 min of MVPA per day. When examined separately by child gender, parents sharing the role of supporting child activity during the week were associated with, on average, an additional 5.9 min of MVPA per day for boys and 0.4 min per day for girls, with no strong statistical evidence of a difference between boys and girls ($P_{interaction}=0.34$). Fathers taking the lead in supporting child activity (compared with mothers) were more weakly associated with child MVPA, with an inverse (rather than positive) association for girls, but again with no strong statistical evidence for gender interaction. Associations for parent support of child physical activity during the weekend showed very similar patterns

**Table 2** Mean difference in the children's average MVPA minutes per day and accelerometer CPM associated with gender of parent who supports physical activity during the week and weekend (n=944)

| Exposure | MVPA (min/day): mean difference (95% CI) | | | P for gender interaction |
|---|---|---|---|---|
| | All (n = 944) | Boys (n = 427) | Girls (n = 517) | |
| Supports child activity during week | | | | |
| Mother (ref) | 0 | 0 | 0 | 0.34 |
| Father | 0.3 (–5.7 to 6.3) | 8.1 (–1.7 to 17.9) | –3.7 (–10.4 to 2.9) | |
| Both parents | 3.5 (0.6 to 6.5) | 5.9 (1.2 to 10.6) | 0.4 (–3.0 to 3.8) | |
| Supports child activity at the weekend | | | | |
| Mother (ref) | 0 | 0 | 0 | 0.22 |
| Father | 1.7 (–2.8 to 6.2) | 5.7 (–1.5 to 12.9) | –3.4 (–8.5 to 1.7) | |
| Both parents | 2.4 (–1.1 to 5.9) | 4.5 (–1.4 to 10.3) | 0.7 (–3.0 to 4.4) | |
| Exposure | Accelerometer CPM: exposure mean difference (95% CI) | | | P for gender interaction |
| | All (n=944) | Boys (n=427) | Girls (n=517) | |
| Supports child activity during week | | | | |
| Mother (ref) | 0 | 0 | 0 | 0.61 |
| Father | 0.7 (–51.7 to 53.2) | 56.7 (–28.8 to 142.1) | –22.8 (–86.7 to 41.1) | |
| Both parents | 28.0 (2.0 to 54.0) | 55.1 (14.3 to 95.9) | 2.8 (–29.9 to 35.4) | |
| Supports child activity at the weekend | | | | |
| Mother (ref) | 0 | 0 | 0 | 0.33 |
| Father | 13.1 (–26.5 to 52.6) | 55.6 (–7.2, to 118.3) | –26.2 (–75.9 to 23.4) | |
| Both parents | 22.6 (–7.7 to 52.9) | 52.8 (1.8 to 103.7) | 4.7 (–31.3 to 40.7) | |

Models are adjusted for child age, parent gender and household IMD score.
CPM, counts per minute; IMD, Index of Multiple Deprivation; MVPA, moderate-to-vigorous physical activity.

to those for weekday activity, but were somewhat weaker in magnitude. In general, the patterns of association with achieving MVPA recommendations were similar to what was found for MVPA as a continuous measure, including point estimates suggesting weaker or inverse effects in girls but no evidence of gender interaction (Table 3). The one exception was that fathers supporting activity at weekends had a similar magnitude of effect as both parents being supporters.

The mean difference in children's CPM by parent/s who supports child activity during the week also showed a similar pattern to that seen for time spent in MVPA (table 2).

## DISCUSSION

The data presented in this paper show that while the participants in this study believe the responsibility of

**Table 3** OR for children achieving 60 min of MVPA per day associated with gender of parent supporting child physical activity during the week and weekend (n=944)

| Exposure | Meeting government guideline: OR (95% CI) | | | P for gender interaction |
|---|---|---|---|---|
| | All (n=944) | Boys (n=427) | Girls (n=517) | |
| Supports child activity during week | | | | |
| Mother (ref) | 0 | 0 | 0 | 0.95 |
| Father | 0.96 (0.54 to 1.72) | 1.61 (0.62 to 4.21) | 0.75 (0.34 to 1.66) | |
| Both parents | 1.60 (1.20 to 2.14) | 2.23 (1.37 to 3.62) | 1.23 (0.83 to 1.82) | |
| Supports child activity at the weekend | | | | |
| Mother (ref) | 0 | 0 | 0 | 0.30 |
| Father | 1.20 (0.78 to 1.86) | 2.10 (1.02 to 4.32) | 0.74 (0.40 to 1.38) | |
| Both parents | 1.20 (0.86 to 1.68) | 1.81 (1.01 to 3.24) | 1.00 (0.64 to 1.54) | |

Models are adjusted for child age, parent gender and household IMD score.
IMD, Index of Multiple Deprivation; MVPA, moderate-to-vigorous physical activity.

supporting child physical activity should be shared between both parents, quantitative data suggest that families mostly share the role on the weekend, with mothers primarily supporting child activity during the week. This finding was mirrored in the interview data, where several mothers reported that they supported child activity during the week, because fathers worked long hours or late into the evening. Despite families traditionally functioning such that one parent (often the mother) takes on more childcare responsibilities in general, it is interesting that parents still feel that supporting child activity should be a shared responsibility. Indeed, traditional familial roles are shifting, and it is now more common for both parents to work and for fathers to take on the role of primary care provider,[54 55] so it may be expected that more fathers are taking an active role in their children's physical activity. We found that the majority of parents reported they shared the role of supporting their child's activity both during the week and at the weekend (40%–65% of mothers and fathers responded this way for both time points; online supplementary table S2).

In quantitative analyses for all three outcomes (time spent in MVPA, meeting MVPA recommendations and CPM), we saw similar patterns of, in general, higher child physical activity where parents reportedly shared the role of supporting their child's physical activity during both weekdays and weekends. For example, both parents supporting child activity equally during the week were associated with boys doing an additional 40 min of MVPA across the week, which could be the difference between a child achieving the recommended guidelines or not. The one exception was for meeting MVPA recommendations at the weekend, where associations of fathers reportedly leading the support were similar to those when both parents shared the responsibility. There was some evidence that positive associations were stronger for sons, and that some associations were inverse for daughters. However, we found no strong statistical evidence that associations differed between sons and daughters, and without further exploration in much larger numbers,

we cannot assume that parental roles in supporting their child's activity differ by the child's gender.

There was some suggestion that mothers were more likely to support their daughter to be active, whereas fathers were more likely to support their son's activity, though caution is needed here given the disparity in which parents provide data, with 72% of families having data from mothers only and 28% from fathers only. Several studies have reported that fathers may be more involved in their son's physical activity[15 31] or have found stronger links between father–son and mother–daughter dyads in terms of their physical activity behaviour.[36–38] In contrast, interview data from the current study revealed a myriad of gender patterns, including examples from fathers supporting girls' physical activity because they were more confident than mothers in supporting physical activity or because they enjoy watching their daughter play football and a mother taking her son mountain biking to engage in quality one-on-one time. There were also examples of fathers taking sons to traditionally male-orientated sports (eg, rugby or football) to bond over shared interests and give mothers a respite from parenting.

The results from the current study suggest that intervention studies should be developed to engage both parents, or specifically fathers, in supporting their children to be active, not necessarily focused on children and parents being active together, but rather on how parents can work together to schedule times for children to be active across the week in both structured and unstructured activities, and how parents can share the role between parenting partners. Table 4 summarises the key findings and implications for how parents can support child activity that have emerged from this study. These suggestions provide ways that researchers and policy-makers can help parents to support their child's physical activity, through providing advice and encouragement to developing family physical activity plans. Research needs to be conducted into how best to operationalise these suggestions and understand the channels that parents typically use for finding parenting advice and ideas for physical activities. Potential

| Table 4 | Key findings and implications for how parents can support their child's physical activity |
|---|---|
| **Finding** | **Implication** |
| Mothers primarily support child physical activity during the week | Develop advice for mothers to help them facilitate their child's physical activity during busy weekdays (eg, identifying times in the day for promoting activity and ideas for active games) |
| Engaging fathers to be involved in supporting child physical activity is important | Encourage fathers to see the important role they can play in supporting their child's activity |
| Children, possibly more so boys, are more active if both parents share the role of supporting child physical activity | Develop family physical activity plans (eg, who can support when) to encourage both parents to take an active role in supporting their child's physical activity |
| Parents can use physical activity time to bond over shared interests or engage in quality one-to-one time with children | Encourage parents to value physical activity time as a way to share interests and bond with children (eg, promote physical activity as quality family time) |
| Some parents, possibly more so mothers, struggle for confidence when it comes to supporting child physical activity | Develop parental skills and confidence in supporting and facilitating child activity and encourage parents to model the behaviours that they wish their child to adopt |

avenues for disseminating advice include encouraging sharing of advice and positive affirmations via parents' peer networks, delivering information through schools or communicating advice via social media and parenting forums.

## Strengths and limitations

A main strength of the study is the mixed-methods approach, using both accelerometer-assessed physical activity from a large sample of children aged 8–9 years and semistructured interview data with parents. This approach provides rich data about the gender roles associated with how parents support their child's activity. Another strength is that we interviewed a relatively large sample of parents, including 20 fathers, a group that are known to be difficult to engage in research.[56] Limitations of the study include its cross-sectional nature so causality could not be examined. In the main dataset, parents were primarily represented by mothers (72%), which is likely to have biased how they responded to questions about who supports their child's activity. In addition, because only one parent was required to participate with their child, this study does not include information on whether children were from same-sex families, single-parent families or where primary caregivers are grandparent or extended family. We had very limited power to explore gender interactions, thus while our results suggest that parent support of their child's physical activity might have a stronger positive impact on sons compared with daughters, it would be wrong to conclude that from these data, and much larger independent studies are required to explore that further. Parental responses to our exposure questions provided no information on the type (quality or quantity) of their supporting role, and thus it is not known whether both parents equally supporting child activity are simply a proxy for greater support. Additionally, the variable ascertaining which parent 'should take the lead in supporting child physical activity' did not differentiate between weekdays and weekend days. A total of 279 families were excluded from the study due to missing data, which may have resulted in sampling bias, because these participants differed from included participants in terms of their MVPA and household IMD score. This study is also drawn from a single UK city area with a primarily white British population, and as such, our ability to extend findings to other settings, countries and ethnicities is limited.

## CONCLUSIONS

We found some evidence that parents share the role of supporting their children to be active. It is possible that mothers primarily support child activity during the week, with the role shared more equally on the weekend. Children are more active when parents share the responsibility of supporting their child's activity, but further large independent studies are required to replicate our findings and determine whether parental support has a stronger

effect on sons than daughters. Future studies should also seek to engage more fathers, verify reports of who takes a supporting role (eg, through cross comparison of reports from each parent and the child or direct observation) and collect information on the nature of supporting roles (quality and frequency).

**Acknowledgements** We would like to thank all of the families and schools that have taken part in the B-PROACT1V project. We would also like to thank all current and previous members of the research team who are not authors on this paper.

**Contributors** Conception/design: RJ, ES-M, JLT, DAL and SJS. Quantitative and qualitative data collection: ES-M. Data analysis/acquisition/interpretation: ES-M, RJ, ZT and DAL. Drafting/revising critically for important content and final approval: all authors. Accountability for study and manuscript: ES-M and RJ.

**Funding** This work was supported by grants from the British Heart Foundation (ref PG/11/51/28986 and SP 14/4/31123). DAL works in a unit that receives funding from the University of Bristol and UK Medical Research Council (MC_UU_1201/5); she is also a UK National Institute of Health and Research Senior Investigator (NF-SI-0166-10196).

**Disclaimer** The funders had no involvement in data analysis, data interpretation or writing of the paper.

**Competing interests** None declared.

**Patient consent** Obtained.

**Ethics approval** The study received ethical approval from the School for Policy Studies Ethics Committee at the University of Bristol.

**Provenance and peer review** Not commissioned; externally peer reviewed.

**Data sharing statement** The datasets generated during the current study are not publicly available as the project is ongoing, and data are not ready for archiving. We will make quantitative data available to the wider research community once the project is completed in August 2019. Because of possible disclosure with qualitative data, we will consider requests to use and further explore those data on a per request basis with an appropriate balance between sharing data as fully as possible while maintaining participant anonymity.

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
