## [Reviewer comments · BMJ Open]

ARTICLE DETAILS

TITLE (PROVISIONAL)	The roles of mothers and fathers in supporting child physical activity: a cross-sectional mixed-methods study
AUTHORS	Solomon-Moore, Emma; Toumpakari, Zoi; Sebire, Simon; Thompson, Janice; Lawlor, Debbie; Jago, Russ

VERSION 1 – REVIEW

REVIEWER	Professor Zoe Knowles Liverpool John Moores University, Liverpool, UK
REVIEW RETURNED	09-Oct-2017

GENERAL COMMENTS	I have enjoyed reading the paper and commend the authors for the quality of the writing and contribution to the field. I am a qualitative researcher and have reviewed with that in mind. Line 34 - state how many were fathers Line 49-52 - why are you stating description when the the premise of thematic analysis is consensus? Line 76 - state telephone interviews Line 108 - amend work to research Line 199-201 - what are the gaps/ findings from previous studies? A summary would be helpful with references cited. 209-226 - 210-211 and 224-226 could be offered as a a few sentences and would be useful at the start of the section. The remainder of the paragraph could be deleted for efficiency Line 257-258 - please offer some detail on accuracy and consistency mentioned here. How were disputes resolved, did any remain unresolved? Line 318 - needs [] with regard to right thing - what does this mean? Line 361-364 - it is authors preferences but leaving repeat words and 'err' in may cause some loss of flow to the reader and/or not translate internationally, whilst I accept this is verbatim the use of ... as regards indicating words removed may be appropriate. Line 448-449 - shouldn't this be 'The data presented in this paper show that whilst the participants in this study believe the responsibility of leading child physical activity should be shared between both parents...' Line 484 - respite maybe as opposed to break Line 495 - this is a useful inclusion and appreciated. I did find myself asking HOW the implications could be operationalised and this may well be ahead for the research team. For example how would this advice be given? I think a few lines extending beyond that of line 494 are warranted. General comment - have the researchers considered those groups that may have same sex parents or where primary caregivers are grandparents/extended family due to circumstances or culture? Do you need to comment on this somewhere in the paper?
---

REVIEWER	Rebecca Gunter York University, Canada
REVIEW RETURNED	18-Oct-2017

GENERAL COMMENTS	Overall I felt that this paper was well written and addressed an interesting area of research regarding potential nuances regarding parent support for child physical activity. The strengths of the paper include using a mixed method approach and accelerometer to measure child MVPA. The key limitations of the paper include a lack of theoretical (or even conceptual) framework to describe the rationale for a possible role for gender in understanding parent support. There is a need to address/consider the potential that "shared responsibility" may largely reflect MORE (i.e., greater quantity) of parent support in general. I also struggled with the concept of leadership. I wasn't entirely clear how this concept was operationalized or why it is important conceptually or theoretically. I have also listed minor considerations in comments within the document. I would be pleased to support the publication of this paper once these revisions are considered. The reviewer also provided a marked copy with additional comments. Please contact the publisher for full details.
--

VERSION 1 – AUTHOR RESPONSE

Reviewer: 1

I have enjoyed reading the paper and commend the authors for the quality of the writing and contribution to the field. I am a qualitative researcher and have reviewed with that in mind.

Response: We thank the reviewer for their supportive comments about the manuscript.

1. Line 34 - state how many were fathers

Response: We have now stated this.

2. Line 49-53 - why are you stating description when the premise of thematic analysis is consensus?

Response: We have now edited this to make it clearer that these were the themes that emerged from the data, as follows: 'Themes emerged from the qualitative data, specifically; parents proactively supporting physical activity equally, mothers supporting during the week, families getting together at weekends, families doing activities separately due to preferences, and parents using activities to bond one-to-one with children.' (line 49-53)

3. Line 77 - state telephone interviews

Response: We have now stated this.

4. Line 109 - amend work to research

Response: We have amended this.

5. Line 199-201 - what are the gaps/ findings from previous studies? A summary would be helpful with references cited.

Response: We have included the gaps identified from previous studies with references cited on lines 220-225.

6. 209-226 - 210-211 and 224-226 could be offered as a few sentences and would be useful at the start of the section. The remainder of the paragraph could be deleted for efficiency

Response: We thank the reviewer for this suggestion, the remainder of the paragraph has been deleted and the two sentences have been incorporated into the start of the methods section (line 149-156).

7. Line 257-258 - please offer some detail on accuracy and consistency mentioned here. How were disputes resolved, did any remain unresolved?

Response: We have now added the following detail (line 260-261): 'Any disagreements that occurred during coding were discussed with additional members of the research team to ensure consensus, and no disagreements remained unsolved.'

8. Line 318 - needs [] with regard to right thing - what does this mean?

Response: In this interview the parent was referring to physical activity as the 'right thing' to do, we have now made this clearer with [physical activity] (line 320).

9. Line 361-364 - it is authors preferences but leaving repeat words and 'err' in may cause some loss of flow to the reader and/or not translate internationally, whilst I accept this is verbatim the use of ... as regards indicating words removed may be appropriate.

Response: We agree with the reviewer's comment and the repeat words and 'err's have now been removed (line 365-367).

10. Line 448-449 - shouldn't this be 'The data presented in this paper show that whilst the participants in this study believe the responsibility of leading child physical activity should be shared between both parents...'

Response: We agree and have made the recommended edit suggested (line 434-435).

11. Line 484 - respite maybe as opposed to break

Response: We have made the suggested edit (line 474).

12. Line 495 - this is a useful inclusion and appreciated. I did find myself asking HOW the implications could be operationalised and this may well be ahead for the research team. For example how would this advice be given? I think a few lines extending beyond that of line 494 are warranted.

Response: We have now added a few lines as follows: 'Research needs to be conducted into how best to operationalise these suggestions and understand the channels that parents typically use for finding parenting advice and ideas for physical activities. Potential avenues for disseminating advice include encouraging sharing of advice and positive affirmations via parents' peer networks, delivering information through schools, or communicating advice via social media and parenting forums.' (line 485-490).

13. General comment - have the researchers considered those groups that may have same sex parents or where primary caregivers are grandparents/extended family due to circumstances or culture? Do you need to comment on this somewhere in the paper?

Response: This is a limitation of this study because we only required one parent to participate with their child, and so we were unable to comment on the proportion of same sex parents or where primary caregivers are from the extended family. We have included a statement to this effect in the limitations section (line 502-505).

Reviewer: 2

Overall I felt that this paper was well written and addressed an interesting area of research regarding potential nuances regarding parent support for child physical activity. The strengths of the paper include using a mixed method approach and accelerometer to measure child MVPA.

Response: We thank the reviewer for their supportive comments about the manuscript.

1. The key limitations of the paper include a lack of theoretical (or even conceptual) framework to describe the rationale for a possible role for gender in understanding parent support.

Response: We have now added a few sentences on the theoretical rationale for the importance of exploring the role of gender in understanding parent support to the Background section (line 114-120).

2. There is a need to address/consider the potential that "shared responsibility" may largely reflect MORE (i.e., greater quantity) of parent support in general.

Response: We agree and we have now made this clearer in the limitations section, as follows: 'Parental responses to our exposure questions provided no information on the type (quality or quantity) of their supporting role, and thus it is not known whether both parents equally supporting child activity is simply a proxy for greater support.' (line 502-505).

3. I also struggled with the concept of leadership. I wasn't entirely clear how this concept was operationalized or why it is important conceptually or theoretically.

Response: We agree with your comments about the concept of leadership being unclear, therefore, in the interests of clarity we have changed the text to refer to parental support (rather than leadership) throughout the manuscript.

I have also listed minor considerations in comments within the document.

4. Title: Title is somewhat inherently miss leading since cross sectional study does not really allow for understanding of directionality??

Response: The title has now been changed to 'The roles of mothers and fathers in supporting child physical activity: a cross-sectional mixed-methods study'.

5. Abstract line 42-44: Is the question regarding parent gender as per the stated objective or child gender as it seems here?

Response: While the primary focus of the study was on the gender of parent who supports child physical activity, because of the disparity in the quantitative results for boys and girls, it seemed pertinent to present the results in the abstract separately for boys and girls (line 42-44).

6. Abstract line 46. Is this meaningful?

Response: Yes, we believe an additional 5.9 minutes per day of MVPA for boys when both parents support child activity during the week is meaningful, because it would correspond to over 40 minutes per week, which could be the difference between a child achieving the recommended guidelines or not. More detail on this has been added to discussion as there was not space to do so in the abstract (line 452 to 455).

7. Line 107-109. Elaborate on the mixed results: unclear what is meant by this statement. How are they mixed? With regard to the value of parental support or with regard to what type of support? Or?

Response: This statement is referring to associations between parent and child physical activity behaviour, rather than support. This sentence has been edited to make this clearer for readers (line 107-109).

8. Line 128. This sentence reads a bit awkward – rephrase to “increased physical activity among children”.

Response: This has now been amended.

9. Line 161. Any comment on validity of wrist-worn accelerometers?

Response: For this study we used waist-worn accelerometers, and so we have added a couple of references that relate to the validity of waist-worn accelerometers in children (Payau et al., 2002; Pate et al., 2006) (Line 162-163).

10. Line 168. I believe there are more recent citations for calculating cut off points with wrist accelerometers in children. Perhaps consider or provide rationale for the method used.

Response: As above, we used waist-worn accelerometers in this study, we have added a rationale for the cut-points chosen on lines 168-171.

11. Line 185. Any other parent demographic information collected? It would be of value to see data regarding parent social economic status, work/employment status, ethnic background etc. These variables are important in understanding physical activity participation and support behaviour.

Response: More detail has been provided in the methods about the Indices of Multiple Deprivation score as an indicator of socio-economic status (lines 190-197). We have also added detail to clarify that parent ethnic origin information was obtained (line 187), and this variable has been included in Table 1 (line 279). We also included the lack of ethnic diversity in the sample as a limitation of the study (line 517-518).

12. Line 201. Did parents wear accelerometers as well? This is not clear above. Also unclear that other biometric data collected.

Response: Yes, parents did wear accelerometers as well, however, this data was not used in the present study, therefore we have tried to eliminate confusion for readers by removing mention of parent accelerometer wear in the methods. A sentence has been added to make it clear that child biometric measures were also measured (line 197).

13. Line 205. Curious why not stratified by meeting versus not meeting the guidelines. While I appreciate that the median is very close to meeting the guidelines, it seems strange to use the median versus the guideline cut off.

Response: The analysis plan for the qualitative interviews was developed at the point of applying for ethical approval before analysis of the quantitative data, hence why the data is stratified around the median, and we wanted to keep the methods consistent for both the physical activity and sedentary time variable (where there is no such definitive threshold), as the stratification was used simply to ensure representativeness in the data.

14. Line 205. Unclear from purpose why sedentary behaviour was measured.

Response: Sedentary behaviour is one of the primary outcomes of interest of the wider longitudinal study, we have now made this clearer on line 144 where we describe the aims of the B-Proact1v study.

15. Line 229. I'm not sure there is value in describing the various types of mixed methods approaches. I would suggest just describing the method used in the study and cite the paper that describes the various approaches.

Response: The detail on the mixed methods approaches has been cut down as recommended and the information retained (with citation) has been incorporated at the start of the Methods section (lines 149-156).

16. Line 270. I assume this [the majority of data being from mothers] is discussed later a limitation.

Response: Yes, the sample is overrepresented by mothers and this may bias how they respond to the support questions. This is mentioned as a limitation of the study (line 500-502).

17. Line 290. It is surprising to me that almost all parents felt that the role should be shared when some/many families may function such that one parent (often the mother) takes on more childcare

responsibilities in general – perhaps worth commenting that despite this system parents still felt it should be a shared responsibility.

Response: We have added a comment on this in the discussion (line 439-442), as follows: ‘Despite families traditionally functioning such that one parent (often the mother) takes on more childcare responsibilities in general, it is interesting that parents still feel that supporting child activity should be a shared responsibility.’

18. Line 293. It might also be interesting to know if children have siblings and if so if siblings are same-sex or different sex as the year 4 child of focus.

Response: This data was collected as part of the wider longitudinal study, however, we feel it is outside the scope of the present study, and has been investigated in previous studies (Edwards et al., 2015; BMJ Open 5:e006593), and so is not discussed further in this paper.

19. Line 333. Interesting that this mother speaks specifically about her sons in this comment but her Year 4 child is a girl. Were parents primed to speak about the year 4 child specifically? If not then that is a limitation that should be mentioned since the MVPA data are specific to the year 4 child and the parents thoughts about support could be reflective of thinking about other/multiple children not just the year 4 child participant.

Response: In the interviews, parents were primed to speak about the Year 4 child specifically, and prompted as such if they strayed off-topic. Therefore, we do not feel it is worth mentioning as a limitation of the study, we considered removing the quote, but feel it is a good example of how parents work together to share the responsibility of supporting their children’s physical activity, and as such have chosen retain it.

20. Line 434. Perhaps a limitation of the study is the lack of distinction in who "should" lead parent support between weekdays and weekends. Perhaps some fathers who think that parents support should be shared area attempting to do so by increasing support on weekends while acknowledging that mothers might play a role in providing more support during the week. Perhaps this mentality explains some of the apparent disconnect between the high levels of parents who believe it should be shared and the relatively low levels of parents who expressed shared weekday responsibilities.

Response: We agree that there is some disconnect between the high levels of parents who believe leadership should be shared and the relatively low levels of parents who expressed shared weekday responsibility, and that your suggestion that fathers may increase support at weekends because they acknowledge mothers provide more support during the week may be a potential reason for this disconnect. However, because weekdays and weekend days were not differentiated between for this variable, it is outside of the scope of the study to make assumptions about the potential causes of this disconnect. Therefore, the fact that the ‘should lead’ variable did not differentiate between weekdays and weekend days has been added as a limitation (line 511-513).

21. Line 505. Could you include effect sizes to increase your confidence about your suspected interactions in the face of limited power?

Response: We have not added effect sizes, as the aim of this paper was to describe associations and not test for the effect of an intervention. We, therefore, do not feel that adding effect sizes would be appropriate.

22. Line 509-510. I would add an explicit note that you do not know the quantity of support and therefore do not know if shared “leadership” simply reflects greater support.

Response: We have tried to make this clearer in the limitations section, as follows: ‘Parental responses to our exposure questions provided no information on the type (quality or quantity) of their supporting role, and thus it is not known whether both parents equally supporting child activity is simply a proxy for greater support’ (line 509-511).

23. Line 515. Was parent MVPA considered in analyses at all?

Response: We did not consider parent MVPA in the current analyses because we have focused on the association between parent and child MVPA behaviour in another study (Jago et al., 2017; IJBNPA 14:110), and as such felt it was outside of the scope of the present study. We have highlighted the existence of this paper in the manuscript.

I would be pleased to support the publication of this paper once these revisions are considered.

Response: Thank you very much for your support.

VERSION 2 – REVIEW

REVIEWER	Professor Zoe Knowles Liverpool John Moores University England, UK
REVIEW RETURNED	30-Oct-2017
GENERAL COMMENTS	Thank you for addressing the comments raised in such a timely manner.